# Beneficial Effect of Exogenously Applied Calcium Chloride on the Anatomy and Fast Chlorophyll Fluorescence in *Rhododendron × pulchrum* Leaves Following Short-Term Heat Stress Treatment

Jianshuang Shen [1,†], Hefeng Cheng [1,†], Xueqin Li [1], Xiangdong Pan [2], Yue Hu [1] and Songheng Jin [1,3,*]

[1] Jiyang College of Zhejiang A&F University, Zhuji 311800, China
[2] Management Office, Wuyanling National Nature Reserve, Wenzhou 325500, China
[3] Department of Life Science and Health, Huzhou College, Huzhou 313000, China
* Correspondence: shjin@zafu.edu.cn
† These authors contributed equally to this work.

**Abstract:** The heat tolerance of plants can be improved by using exogenous calcium chloride ($CaCl_2$) to cope with temperature fluctuations. Since global climates continue to warm, it is important to further explore the way in which plants respond to heat stress with the use of $CaCl_2$. We aimed to explore the effect of exogenous $CaCl_2$ on the leaf microstructure, leaf epidermal ultrastructure, and chlorophyll a fluorescence of *Rhododendron × pulchrum* (*R. × pulchrum*) under heat stress. In the leaves of *R. × pulchrum* treated with exogenous $CaCl_2$, compared to the control, the thickness of the epidermis, spongy tissues, and stomatal aperture increased, whereas the stomata density and ratio of closed/open stomata decreased. In the leaves of *R. × pulchrum* under heat stress conditions, compared to the control, the values of the maximal photochemical efficiency of photosystem II ($F_v/F_m$), the performance index on an absorption basis ($PI_{ABS}$), the quantum yield for the reduction of terminal electron acceptors on the acceptor side of PSI ($\varphi_{Ro}$), and the energy absorbed per unit cross-section of a photosynthesizing object at the moment of achieving the fluorescence maximum ($ABS/CS_M$) all decreased, whereas the quantum yield of the energy dissipation ($\varphi_{Do}$) increased significantly. However, these differences disappeared when *R. × pulchrum* was treated with exogenous $CaCl_2$. This suggests that exogenous $CaCl_2$ can improve the heat tolerance in *R. × pulchrum* by regulating the leaf anatomical structure and the behavior of epidermal cells and stomata in leaves, protecting the stability of photosystems I and II and improving the electron transfer from $Q_A$ to $Q_B$. Our study could provide a theoretical basis for the breeding, further research, and utilization of Rhododendron in the context of global warming.

**Keywords:** heat stress; *Rhododendron*; exogenous calcium chloride; chlorophyll fluorescence; anatomy

## 1. Introduction

Global climates continue to warm, accompanied by an increase in $CO_2$ concentration and sustained elevated temperatures [1]. As a result, green plants are negatively affected and will have to adapt to high temperatures in the future [2]. Green plants have the capacity to cope with temperature fluctuations; however, this capacity might not keep pace with global warming. The heat tolerance of plants can be improved using different methods, including genetic selection and the application of exogenous regulators, such as calcium chloride ($CaCl_2$) and salicylic acid. These methods aid in physiological response adaptation in plants [3,4].

In model plant and crop studies, it has been shown that exogenous $CaCl_2$ can improve heat resistance and enable adaptation to higher temperatures [5–7]. $Ca^{2+}$, as a secondary messenger, regulates physiological and biochemical mechanisms in plants adapted to

adverse environmental conditions. $Ca^{2+}$ participates in the regulation of stomatal movements [3] and gene expression [8], influencing the chlorophyll content and antioxidant enzyme activity [9].

High-temperature stress can damage the homeostasis of the membrane system in plants and affect their photosynthetic energy metabolism and physiological and biochemical processes. Under heat stress, various metabolic activities in plants are influenced, such as their effects on the leaf phenotype, anatomy, physiology, and photosynthesis capability [10–14]. Previous studies have indicated that green plants are characterized by heat tolerance derived from their leaf anatomical and ultrastructural traits, including a decrease in thick leaves, the ratio of palisade to spongy parenchyma, and epidermis cell size, as well as stomatal closure and an increase in stomatal density [15–17]. Few studies have reported the effect of exogenous $CaCl_2$ on the plant leaf structure.

The photosynthetic apparatus is one of the most sensitive components affected by heat stress [18,19]. The efficiency of the photosynthetic apparatus can affect chlorophyll fluorescence parameters and gas exchange, reflecting the photosynthetic electron transfer (light reaction) and photosynthetic rate (dark reaction), respectively [20,21]. In recent years, the measurement of chlorophyll fluorescence has been used to investigate photosynthetic characteristics in plant leaves under heat stress, including delayed chlorophyll a fluorescence (DF), prompted chlorophyll a fluorescence (PF), and modulated reflection at 820 nm (MR) [22]. Chlorophyll fluorescence parameters reflect the electron transfer in photosystem I (PSI), photosystem II (PSII), and photosynthetic electron carriers in the photosystem [23]. The measurement of chlorophyll fluorescence has been widely used to investigate changes in the leaves of plants under heat stress, such as tomato [24], wheat [25,26], barley [27], weeds [28], and *Pinellia ternata* [29]. In recent years, simultaneous measurements of DF, PF, and MR kinetics have been introduced to investigate the response of plants to environmental stresses, such as heat [22] and NaCl [30,31], used in the fields of agronomy, plant science, and environmental science [32,33]. For example, the PF kinetics showed an O-J-I-P curve (O, at 20 μs; J, at 2 ms; I, at 30 ms; P, equal to FM), which is used to quantify the architecture and behavior of PSII. In DF kinetics, parameters $I_1$ (7 ms) and $I_2$ (50 ms) indicate the electron transfer capacity of PSII and the re-oxidation of $PQH_2$ by PSI reaction center ($P_{700}$). MR/MR0 kinetics, usually as an indicator of the redox state changes in plastocyanin and $P_{700}$ [32,33]. This method is important for investigating changes in the photosynthetic electron transfer chain, including the PSI electron acceptor side, electron transport between PSII and PSI, and the PSII electron donor side [22,23,34]. To date, little research has been conducted with respect to the effect of heat stresses on chlorophyll a fluorescence in the leaves of the *Rhododendron* species [35,36].

*Rhododendron* is one of the most popular ornamental plants worldwide, with excellent ornamental traits. The genus *Rhododendron* belongs to Ericaceae and contains more than 1000 species and 25,000 hybrids that are classified into eight subgenera [37,38]. A previous study constructed a phylogeny tree of 200 *Rhododendron* species according to the nuclear and chloroplast genes [39]. *Rhododendron* species are mostly distributed and native to high-altitude areas, growing at elevations between 1000 and 5000 m [40]. Many *Rhododendron* species are seriously affected by high temperatures in low-altitude areas, resulting in a decline in ornamental quality. Global warming is an emerging major challenge to the growth and development of *Rhododendron* species [9]. Thus, the heat resistance of *Rhododendron* species needs to be improved to face global challenges.

To date, only a limited number of studies have investigated the heat resistance of *Rhododendron* in response to exogenous $CaCl_2$. Previous studies have suggested that exogenous $CaCl_2$ has a positive effect on the plant growth, chlorophyll content, total soluble protein levels, and enzymatic antioxidant activity under heat stress [9,16]. However, investigations of the response of *Rhododendron* to heat stress when exposed to exogenous $CaCl_2$ from anatomical and photosynthesis perspectives are scarce. *R.* × *pulchrum* Sweet, belonging to Subg. Tsutsusi of *Rhododendron*, is severely affected by high temperatures, appearing to grow normally or even die under high temperatures. In this study, we

hypothesized that exogenous $CaCl_2$ could aid the heat tolerance of *R. × pulchrum*, and we explored the effect of exogenous $CaCl_2$ on the leaf microstructure, leaf epidermal ultrastructure, and chlorophyll a fluorescence of *R. × pulchrum* under heat stress. Our study could provide a theoretical basis for the breeding, further research, and utilization of *Rhododendron* to assist in sustaining forestry development [41].

## 2. Materials and Methods

### 2.1. Plant Material

Two-year-old cutting plants of *R. × pulchrum* were obtained from a single tree grown at the Jiyang College of Zhejiang A&F University, Zhejiang, China (29°44′51″ N, 120°15′17″ E). These cutting trees were planted in pots (15 cm tall, 20 cm top diameter) individually. Each pot was filled with 5 kg loam soil (field water-holding capacity of 33%) and watered when the pot was naturally dried (watered to saturation each time, approximately 0.3 L per pot). Six pots were randomly selected per treatment. Using the foliar sprays method, the different concentrations of $CaCl_2$ were foliar sprayed with approximately 10 mL per plant, evenly spraying both sides of the leaves. The experiment set different concentrations of $CaCl_2$ (0, 5, 10, 20, 30 mmol/L) based on similar studies. The results showed that 10 mmol/L was the best treatment. The heat resistance capacity can be significantly improved compared with the control group. Thus, four treatments were set up as follows: control treatment with distilled water (CK + W), control treatment with 10 mM $CaCl_2$ (CK + $Ca^{2+}$), heat stress treatment with distilled water (H + W), and heat stress treatment with 10 mM $CaCl_2$ (H + $Ca^{2+}$).

The experiment was carried out in a growth chamber at the Jiyang College of Zhejiang A&F University. The temperature within the chamber was 28 °C/25 °C (day/night), the photoperiod was 14/10 h (day/night), the light intensity was 600 $\mu mol \cdot m^{-2} \cdot s^{-1}$, and the humidity was 70 ± 5%. An aqueous solution of 10 mM $CaCl_2$ (Yudinghuagong, Shandong, China) was applied daily at 5 p.m. using the foliar sprays method. The control treatment involved only water, and tests were conducted for 7 days. Then, the temperature within the chamber of heat stress treatments was changed to 40/30 °C (day/night), keeping soil water content stable during the experiment; the other conditions remained under the original conditions. Measurements were recorded 2 days after heat stress treatments.

### 2.2. Microstructure of the Leaves and Ultrastructure of Leaf Epidermal Cells

Mature leaves, selected from the third fully expanded leaf, were collected from the plants and cut into 2 mm × 4 mm pieces; dehydrated pieces were then obtained for subsequent experiments. To observe the microstructure of the leaves, semi-thin sections were made following Zhang's method [42]. The dehydrated samples were embedded in epoxy resin and sliced with a LEICA DC6 ultramicrotome (Leica Microsystems, Wetzlar, Germany). Ultra-thin sections (1 μm thick) were cut following Ruppel's method [43]. The sections were stained with toluidine blue [15,42] and observed under a microscope. To observe the ultrastructure of leaf epidermal cells, the samples were subjected to freeze-drying (K-850, Emitech Ltd., Kent, UK), after which, they were sprayed with gold particles (E-1010, Hitachi, Tokyo, Japan). The leaf epidermal cells were observed via scanning electron microscope (5136, TESCAN, Brno, Czech Republic). Three different vision fields under the microscope were randomly chosen, and each vision field was repeated three times. The size of the leaf tissues and leaf epidermal stomatal diameter were measured using ImageJ v1.8.0 software. Thirty cells and stomatal were measured and the ratio of palisade thickness to leaf thickness, the ratio of palisade to spongy parenchyma thickness, and stomatal aperture were calculated [44]. In addition, the number of stomata and cells per unit area was counted [45].

### 2.3. Measurement of PF, DF and MR Kinetics

The third and fourth leaves from the top of the shoots were used to determine the PF, DF, and MR kinetics simultaneously after 30 min of dark adaptation using a multi-function

plant efficiency analyzer (PE304NE, M-PEA, Hansatech, Norfolk, UK). The PF, DF, and MR kinetic curves were plotted according to the methods described in previous studies [46–48]. Parameters were derived from chlorophyll a fluorescence [46,47]. Three replicates per treatment were randomly selected.

### 2.4. Statistical Analysis

The experimental design was a completely randomized design. The data were analyzed using SPSS 18.0 software (SPSS Inc. Chicago, IL, USA), and one-way ANOVA and Duncan's multiple range test were performed for the data variable, with the significance level set at the default of $\alpha = 0.05$.

### 3. Results

#### 3.1. Phenotype and Microstructure of the Leaf

The phenotype of the plants under the four treatments is shown in Figure 1. There were significant differences among CK + W, CK + Ca$^{2+}$, and H + Ca$^{2+}$, whereas the leaves presented wilting with a burnt edge in the H + W treatment. Significant differences in the mesophyll tissue thickness were observed among the four groups in this study. As shown in Figure 2 and Table 1, in the H + W treatment, the thickness of the upper epidermis and the ratio of palisade to spongy parenchyma in the leaves were both significantly decreased when compared to CK + W, whereas the thickness of the palisade parenchyma, spongy parenchyma, lower epidermis, leaf, and ratio of palisade to leaf in the leaves were not significantly different. In the CK + Ca$^{2+}$ treatment, the leaf thicknesses of the upper epidermis, spongy parenchyma, and lower epidermis were all significantly increased compared to the control treatment with distilled water. The palisade thickness was not significantly different between CK + Ca$^{2+}$ and CK + W, whereas the ratio of palisade to leaf tissue and the ratio of palisade to spongy parenchyma in CK + Ca$^{2+}$ were significantly lower than CK + W. In the H + Ca$^{2+}$ treatment, the thickness of the upper epidermis, palisade parenchyma, lower epidermis, ratio of palisade to spongy parenchyma, and ratio of palisade to leaf were significantly decreased compared to CK + Ca$^{2+}$. In CK + Ca$^{2+}$ and H + Ca$^{2+}$, the leaf thickness was significantly increased compared to H + W and CK + W.

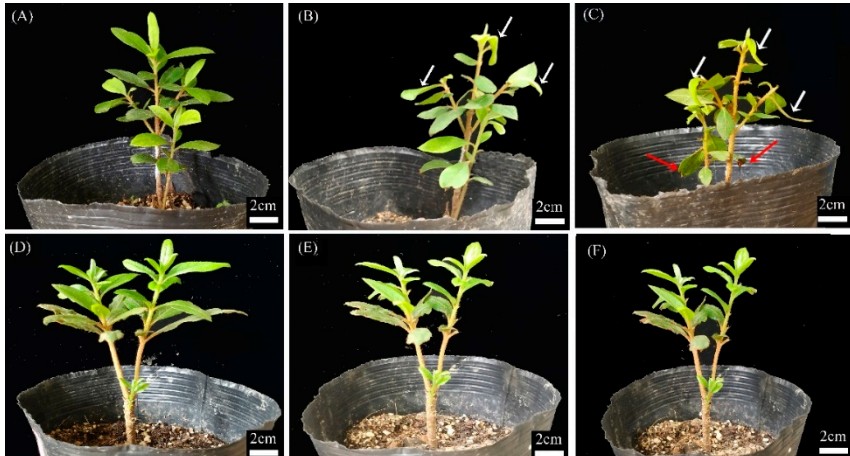

**Figure 1.** The phenotypes of the plants under the four treatments. (**A**) CK + W represents the control treatment with distilled water; (**B**) H + W represents the heat stress treatment after 1 day with distilled water; (**C**) H + W represents the heat stress treatment after 2 days with distilled water; (**D**) CK + Ca$^{2+}$ represents the control treatment with 10 mM CaCl$_2$; (**E**) H + Ca$^{2+}$ represents the heat stress treatment after 1 day with 10 mM CaCl$_2$; (**F**) H + Ca$^{2+}$ represents the heat stress treatment after 2 days with 10 mM CaCl$_2$. Red arrow points to the leaves with burnt edge; white arrows point to the wilting leaves.

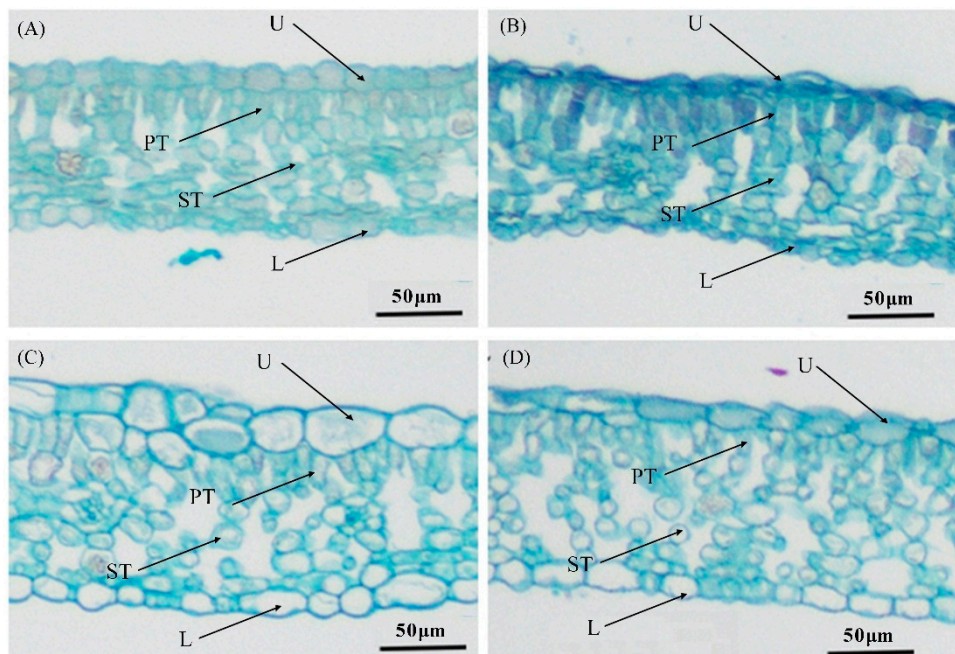

**Figure 2.** Leaf anatomical structure of *Rhododendron* × *pulchrum* leaves under four treatments. (**A**) CK + W represents the control treatment with distilled water; (**B**) H + W represents the heat stress treatment with distilled water; (**C**) CK + Ca$^{2+}$ represents the control treatment with 10 mM CaCl$_2$; (**D**) H + Ca$^{2+}$ represents the heat stress treatment with 10 mM CaCl$_2$. L = lower epidermis; PT = palisade tissue; ST = spongy tissue; and U = upper epidermis.

**Table 1.** Leaf anatomical structure of *Rhododendron* × *pulchrum* leaves under the four treatments.

| Group | Upper Epidermis Thickness (μm) | Palisade Thickness (μm) | Spongy Thickness (μm) | Lower Epidermis (μm) | Leaf Thickness (μm) | Ratio of Palisade Thickness to Spongy Thickness | Ratio of Palisade Thickness to Leaf Thickness |
|---|---|---|---|---|---|---|---|
| CK + W | 14.96 ± 1.79 b [1] | 33.47 ± 2.51 a | 39.58 ± 1.76 c | 12.27 ± 1.39 b | 100.28 ± 3.23 b | 0.85 ± 0.09 a | 0.33 ± 0.02 a |
| H + W | 11.82 ± 1.70 c | 32.53 ± 3.53 a | 43.69 ± 6.36 c | 10.78 ± 2.25 b | 98.82 ± 9.84 b | 0.75 ± 0.08 b | 0.33 ± 0.02 a |
| CK + Ca$^{2+}$ | 22.62 ± 3.43 a | 31.81 ± 4.25 a | 53.78 ± 3.32 b | 17.29 ± 3.31 a | 125.53 ± 5.36 a | 0.59 ± 0.10 c | 0.25 ± 0.03 b |
| H + Ca$^{2+}$ | 15.40 ± 1.53 b | 22.67 ± 6.17 b | 65.91 ± 3.13 a | 13.25 ± 2.90 b | 117.23 ± 6.80 a | 0.34 ± 0.09 d | 0.19 ± 0.04 c |

[1] Mean ± SE followed by the same letter in the same column indicates no significant difference at the 0.05 level based on Duncan's test.

### 3.2. Ultrastructure of Leaf Epidermal Cells and Stomata

The ultrastructure of the leaf epidermal cells and stomata was observed and is shown in Table 2 and Figure 3. Regarding the cell density in the upper epidermis, significant differences were observed among the four treatments, with the order from high to low being (H + Ca$^{2+}$) > (CK + W) > (CK + Ca$^{2+}$) > (H + W). In the leaves of plants in CK + Ca$^{2+}$ and H + Ca$^{2+}$, the stomatal density was significantly lower than that in CK + W, whereas it was significantly higher than that in H + W. In the leaves of plants in CK + Ca$^{2+}$ and H + Ca$^{2+}$, the stomatal aperture was significantly increased compared to that in the other treatments with distilled water (H + W and CK + W). In the leaves of plants in the CK + Ca$^{2+}$ and H + Ca$^{2+}$ treatments, the closed/open stomatal ratios were significantly lower than that in the control treatment (H + W and CK + W). In H + W, the closed/open stomatal ratio in the leaves was significantly increased compared to that in CK + W.

**Table 2.** Ultrastructure of leaf epidermal cells of *Rhododendron × pulchrum* leaves under the four treatments.

| Group | Cell Density (No. of Upper Epidermises/mm²) | Stomatal Density (No. of Stomata/mm²) | Stomatal Transverse Diameter/μm | Stomatal Longitudinal Diameter/μm | Stomatal Aperture/μm² | Open Stomatal Density (No. of Opened Stomata/mm²) | Closed Stomatal Density (No. of Closed Stomata/mm²) | Ratio of Closed/Open Stomata |
|---|---|---|---|---|---|---|---|---|
| CK + W | 217 ± 5 b [1] | 71 ± 3 a | 5.56 ± 0.46 b | 1.43 ± 0.21 b | 6.23 ± 0.99 b | 57 ± 5 a | 14 ± 3 a | 0.25 ± 0.06 b |
| H + W | 81 ± 6 d | 27 ± 3 c | 4.44 ± 0.71 c | 1.44 ± 0.27 b | 4.95 ± 0.78 b | 19 ± 3 c | 8 ± 1 b | 0.43 ± 0.07 a |
| CK + Ca²⁺ | 104 ± 8 c | 46 ± 6 b | 7.38 ± 1.23 a | 2.31 ± 0.37 a | 13.55 ± 3.85 a | 45 ± 5 b | 1 ± 1 c | 0.03 ± 0.01 c |
| H + Ca²⁺ | 251 ± 8 a | 57 ± 5 b | 8.02 ± 0.56 a | 2.39 ± 0.24 a | 15.03 ± 1.71 a | 57 ± 4 a | 1 ± 0 c | 0.02 ± 0.00 c |

[1] Mean ± SE followed by the same letter in the same column indicates no significant difference at the 0.05 level based on Duncan's test.

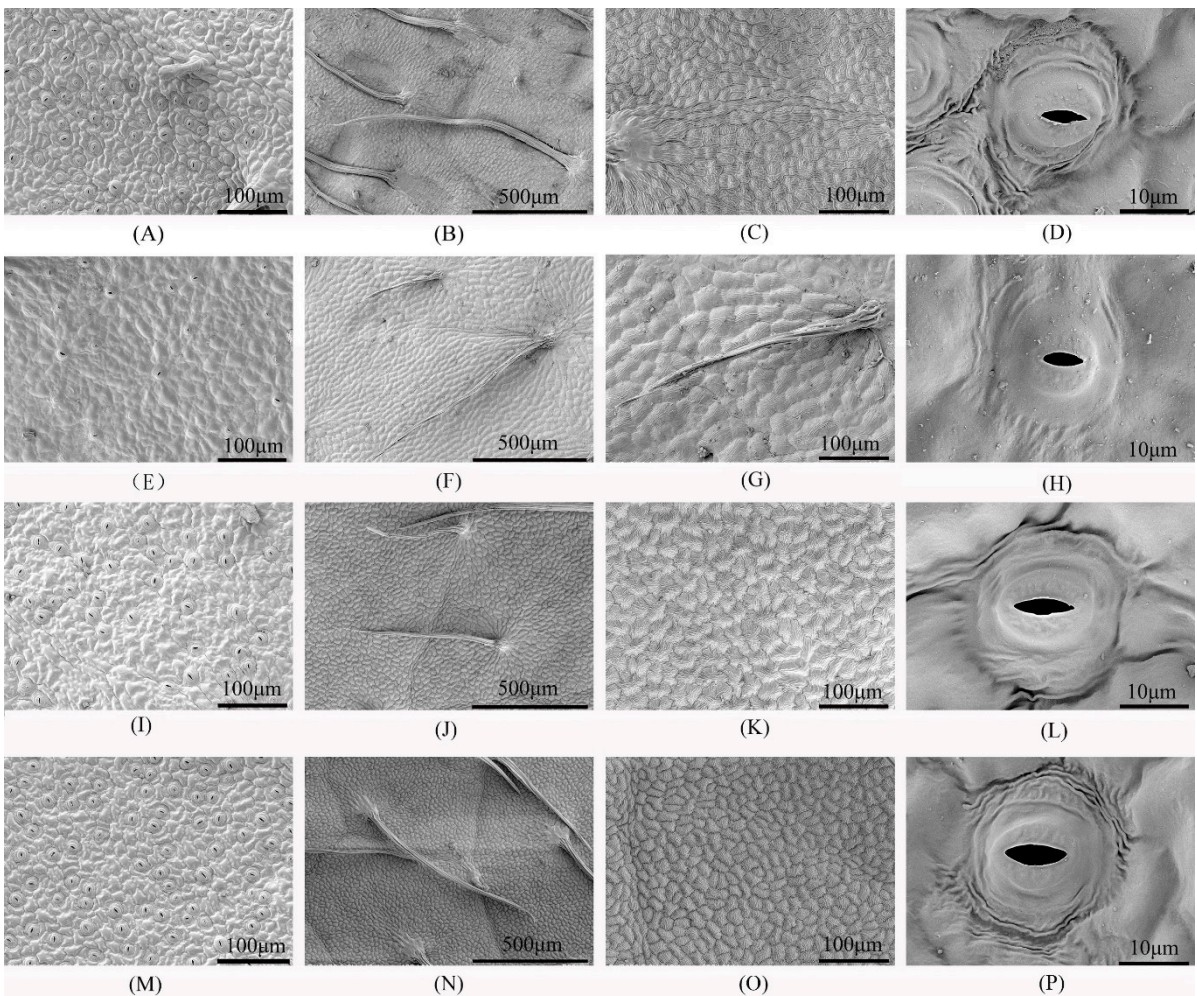

**Figure 3.** Ultrastructure of leaf epidermal cells of *Rhododendron × pulchrum* leaves under four treatments. (**A–D**) CK + W represents the control treatment with distilled water; (**E–H**) H + W represents the heat stress treatment with distilled water; (**I–L**) CK + Ca²⁺ represents the control treatment with 10 mM CaCl₂; (**M–P**) H + Ca²⁺ represents the heat stress treatment with 10 mM CaCl₂; (**A,E,I,M**) show the stomatal density in the lower epidermis; (**B,C,F,G,J,K,N,O**) show the cell density in the upper epidermis; (**D,H,L,P**) show the stomatal aperture in the lower epidermis.

### 3.3. Analysis of PF, DF and MR Kinetics

Figure 4A shows that the chlorophyll-fluorescence-induced dynamic (OJIP) curve of the *R. × pulchrum* leaves of the H + W treatment decreased compared to those in the other three treatments at relative fluorescence intensities $F_I$ of point I and $F_P$ of point P. The $\Delta_{Vt}$ curve showed (Figure 4B) that the ΔK, ΔJ, and ΔI of the H + W treatment were higher than

those of the other three treatments. In DF induction kinetics (Figure 4C), the fast phase occurred until 250 ms and only includes the I (7 ms) peak. Meanwhile, the I value decreased in the *R. × pulchrum* leaves of the H + W treatments, whereas it increased in the leaves of the CK + $Ca^{2+}$ and H + $Ca^{2+}$ treatments. The shape of the $MR/MR_O$ kinetics in the leaves of H + W was more obvious than in the other three treatments (Figure 4D).

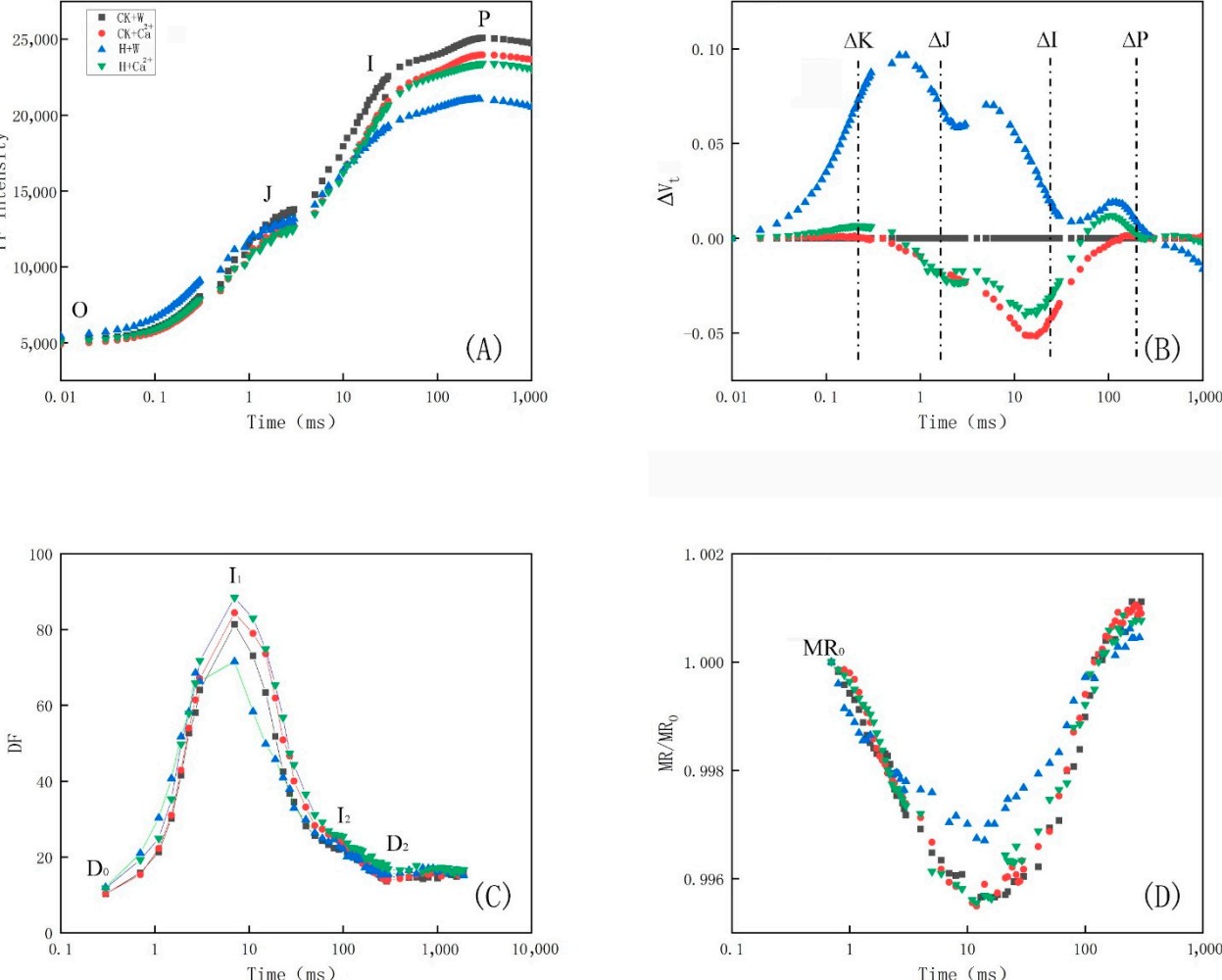

**Figure 4.** PF, DF, and MR kinetics in the leaves of *Rhododendron × pulchrum* under four treatments. Each curve is the average of three replicates. (**A**) Prompt chlorophyll a fluorescence (PF). The signals were plotted on a logarithmic time scale. The signals were fluorescence intensity at 20 μs ≡ $F_O$, 3 ms ≡ $F_J$, and 30 ms ≡ $F_I$; maximum fluorescence intensity, $F_P = F_M$. (**B**) Effect of the development of PSII on the relative variable fluorescence [$\Delta_{Vt} = (F_t − F_0)/F_v − V_{t\,(CK + W)}$]; for $\Delta_{Vt}$ analysis, the fluorescence of the leaves of the control treatment with distilled water (CK + W) was used as the reference and equaled 0. (**C**) Delayed chlorophyll a fluorescence (DF). The signals were plotted on a logarithmic time scale. (**D**) Modulated 820 nm reflection ($MR/MR_O$). The signals were plotted on a logarithmic time scale.

The PF parameters for the *R. × pulchrum* leaves of the four treatments were influenced, as shown in Table 3. There were no significant differences in parameters $\psi_{Eo}$, $\varphi_{Eo}$, and $\delta_{Ro}$ among the four treatments. However, the parameters $F_v/F_m$, $PI_{ABS}$, $\gamma_{RC}$, $\varphi_{Ro}$, and $\varphi_{Do}$ were obviously influenced. Among these parameters, only $\varphi_{Ro}$ had significant changes between the two control treatments (CK + $Ca^{2+}$ and CK + W). Furthermore, the values of $\varphi_{Ro}$ increased in the *R. × pulchrum* leaves treated with $CaCl_2$. The values of $F_v/F_m$, $PI_{ABS}$, and $\gamma_{RC}$ in the *R. × pulchrum* leaves of the heat stress treatment with distilled water were significantly decreased compared to the other three treatments. The values of $\varphi_{Do}$

significantly increased in the leaves of H + W compared to CK + W, whereas there was no significant difference between the two treatments with $CaCl_2$ (CK + $Ca^{2+}$ and H + $Ca^{2+}$).

**Table 3.** Parameters derived from prompt chlorophyll a fluorescence transients in the leaves of *Rhododendron × pulchrum* under the four treatments.

| Group | $F_v/Fm$ [1] | $PI_{ABS}$ [2] | $\Psi_{Eo}$ [3] | $\sigma_{Ro}$ [4] | $\gamma_{RC}$ [5] | $\varphi_{Eo}$ [6] | $\varphi_{Ro}$ [7] | $\varphi_{Do}$ [8] |
|---|---|---|---|---|---|---|---|---|
| CK + W | 0.80 ± 0.01 a [9] | 5.16 ± 0.44 a | 0.61 ± 0.06 a | 0.20 ± 0.04 a | 0.38 ± 0.03 a | 0.49 ± 0.06 a | 0.10 ± 0.02 bc | 0.20 ± 0.01 b |
| CK + $Ca^{2+}$ | 0.80 ± 0.02 a | 5.44 ± 0.16 a | 0.63 ± 0.07 a | 0.25 ± 0.02 a | 0.37 ± 0.03 ab | 0.50 ± 0.06 a | 0.13 ± 0.01 a | 0.20 ± 0.02 b |
| H + W | 0.75 ± 0.03 b | 2.57 ± 0.47 b | 0.55 ± 0.12 a | 0.22 ± 0.09 a | 0.30 ± 0.04 b | 0.41 ± 0.10 a | 0.08 ± 0.02 c | 0.25 ± 0.03 a |
| H + $Ca^{2+}$ | 0.79 ± 0.02 ab | 4.72 ± 0.97 a | 0.63 ± 0.04 a | 0.24 ± 0.02 a | 0.36 ± 0.04 ab | 0.50 ± 0.05 a | 0.12 ± 0.00 ab | 0.21 ± 0.02 ab |

[1] $F_v/F_m$, maximal quantum efficiency of PSII; [2] $PI_{ABS}$, the performance index on an absorption basis; [3] $\psi_{Eo}$, a trapped exciton moves an electron into the electron transport chain beyond $Q_A^-$ at t = $F_o$; [4] $\delta_{Ro}$, efficiency required for an electron to be transferred from reduced carriers between the two photosystems to the PSI end acceptors. [5] $\gamma_{RC}$, a given chlorophyll *a* molecule that functions as the PSII reaction center; [6] $\varphi_{Eo}$, quantum efficiency of electron transfer at t = $F_o$; [7] $\varphi_{Ro}$, quantum yield for reduction of the terminal electron acceptors on the PSI acceptor side; [8] $\varphi_{Do}$, quantum yield for energy dissipation at t = $F_o$; [9] mean ± SE followed by the same letter in the same column indicates no significant difference at the 0.05 level, based on Duncan's test.

The absorbed energy fluxes of the two models are shown in Figure 5. In the membrane model (Figure 5A), the ABS/RC, $TR_0$/RC, and $DI_0$/RC values were all significantly increased in the leaves of H + W compared to CK + W, whereas these three parameters showed no significant difference in the leaves between the two treatments with $CaCl_2$ (CK + $Ca^{2+}$ and H + $Ca^{2+}$). The values of $ET_0$/RC were not significantly different among the four treatments. In the leaf model (Figure 5B), $RC/CS_M$, $ABS/CS_M$, $TR_0/CS_M$, and $ET_0/CS_M$ were all decreased, and $DI_0/CS_M$ was increased significantly in the leaves of H + W compared to CK + W, whereas these parameters had the same trend but were not significantly different in the leaves of H + $Ca^{2+}$ compared to CK + $Ca^{2+}$.

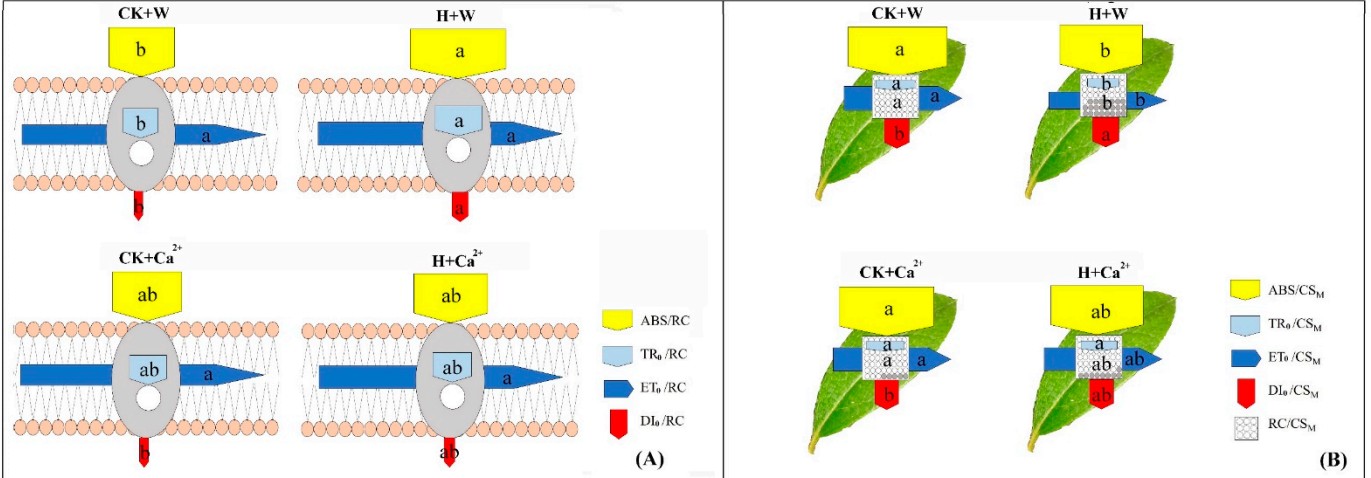

**Figure 5.** Energy flux models in photosystem II for the leaf samples of *Rhododendron × pulchrum* under four treatments. The relative value of each parameter is proportional to the width of the arrows. (**A**) Membrane model represents the specific activity on a single RC basis. The average size of the antenna is represented by the ABS/RC parameter. The fluxes of absorbed energy that were trapped by nonreducing PSII centers are shown as diagonally hatched areas in ABS/RC and TR/RC contours. (**B**) Leaf model represents the phenomenological parameters normalized per unit cross-section (CS) of a photosynthesizing object. The $Q_A$-nonreducing reaction centers are shown with black circles, and the $Q_A$-reducing active centers are depicted with white circles. These were determined at minimum (index 0) or maximum (index m) levels of chlorophyll fluorescence; for example, $ET_0$ or $ET_m$. The fluxes presented in the scheme were normalized to the maximal fluorescence level (m). The same letters indicate no significant difference at the 0.05 level based on Duncan's test.

## 4. Discussion

### 4.1. Exogenous CaCl2 Influences Leaf Structure under Heat Stress

Significant differences in the ratio of palisade to spongy parenchyma in the leaves of *R. × pulchrum* between CK + W and H + W were found in this study (Table 1, Figure 2). This phenomenon was consistent with a previous study on grapevine [15]. After the use of exogenous $CaCl_2$, the thickness of the epidermis and spongy tissues both significantly increased in the leaves of *R. × pulchrum* compared to the control under normal temperatures. In the heat stress treatment with $CaCl_2$, the thickness of the upper epidermis, palisade parenchyma, and lower epidermis, the ratio of palisade to spongy parenchym, and the ratio of palisade to leaves were all significantly decreased when compared to the control treatment with $CaCl_2$. Previous studies have reported that the larger the ratio of palisade to spongy tissues in plant leaves, the stronger the heat tolerance [49,50]. Thus, we predicted that exogenous $CaCl_2$ would improve plant heat resistance, which requires further study. There may be other strategies that differ from the existing research.

In general, cell size reduction, stomatal closure, and an increased stomatal density enable plants to cope with heat stress [51]. This study found that the cell density decreased, the stomatal density decreased, the stomatal aperture decreased, and the ratio of closed/open stomata increased in the leaves of *R. × pulchrum* in response to heat stress (Table 2, Figure 3). The behavior of stomata in the leaves of *R. × pulchrum* under heat stress could curtail water loss and affect gas exchange and instantaneous water-use efficiency; this result is consistent with a previous study [16,51]. The present study found that the cell density and thickness of the upper epidermis both decreased in the leaves of *R. × pulchrum* under heat stress.

The mesophylic tissue consists of spongy and palisade tissue in the *R. × pulchrum*. The palisade tissue is usually continuous, with 1~2 rows in most plants [15,52]. Palisade tissue in the leaves of *R. × pulchrum* is continuous and has more than one row; however, it became discontinuous after the exogenous application of calcium chloride under heat stress treatment. Further, the morphology and thickness of the palisade and spongy tissues of *R. × pulchrum* were almost unaffected by heat stress. This might be a mechanism developed by *R. × pulchrum* to adapt to heat stress and protect mesophyll cells. However, a study of field-grown grapevines found no relationship among water-use efficiency, heat stress tolerance, and stomatal traits [53]. The larger the stomatal density in plant leaves, the stronger the heat tolerance [49,50]. Thus, we predicted that different plant species may have their own strategies to cope with heat stress, including cell and stomatal behavior.

The ability of plants to adapt to the external environment can be improved with the use of exogenous regulators, such as $CaCl_2$, salicylic acid (SA), and putrescine. The spraying of exogenous regulators could ameliorate the plant's adaptation of physiological responses to environmental stresses on its growth and physiological and anatomical features [3,4,45,54]. Previous studies have demonstrated that exogenous $CaCl_2$ can improve heat resistance of plants and enable them to adapt to higher temperatures [5–7]. However, the effect of exogenous $CaCl_2$ on plant leaf epidermal cells and stomatal behavior remains to be further studied. After the use of exogenous $CaCl_2$, the cell density decreased, the stomatal density decreased, the stomatal aperture increased, and the ratio of closed/open stomata decreased in the leaves of *R. × pulchrum* under normal temperatures. However, the morphology and behavior of stomata in the leaves of *R. × pulchrum* treated with exogenous $CaCl_2$ showed no significant change; but the cell density increased and the ratio of closed/open stomata decreased in the H + $Ca^{2+}$ leaves compared with H + W, illustrating that the heat stress was moderate and not severe and most of the stomata in the H + $Ca^{2+}$ leaves appear to be open (Figure 3M and Table 3). This phynomenon maybe through transpiration, accelerates water transport and increases the heat tolerance of plants. These results suggest that exogenous $CaCl_2$ improves the heat tolerance of *R. × pulchrum* by regulating the behavior of epidermal cells and stomata in leaves. The mechanism that $CaCl_2$ protect against heat stress by regulating the behavior of epidermal cells and stomata in leaves need to be further studied.

*4.2. Exogenous CaCl₂ Influences Leaf Chlorophyll Fluorescence in R. × pulchrum under Heat Stress*

In our study, chlorophyll a fluorescence was used to study the photosynthetic energy and electron transfer in *Rhododendron* plants in response to heat stress. The OJIP curve of the *R. × pulchrum* leaves under the heat stress treatment with distilled water (H + W) decreased compared to H + $Ca^{2+}$ treatment at relative fluorescence intensities $F_I$ of point I and $F_P$ of point P (Figure 4A); these results are consistent with previous studies [25,53]. Our research confirmed that OJIP transient was a reliable method for the detection of heat stress in PS II of *Rhododendron* [55]. The shape of the MR/$MR_O$ kinetics in the leaves of H + W was obviously different from CK + W and H + $Ca^{2+}$ (Figure 4D). The fast phase and slow phase of MR/$MR_O$ kinetics in the leaves of H + W were lower than the other three treatments, and the lowest point of the MR/$MR_O$ kinetics in the leaves of H + W was higher than the other three treatments, illustrating that the oxidation and re-reduction abilities of the PS I reaction center and PC were reduced [31]. Our results indicate that the kinetics of the PF and MR of *R. × pulchrum* respond to heat stress, whereas these responses disappear when using exogenous CaCl₂. A previous study reported that a decrease in the signal point of OJIP may be related to a reduction in the antenna pigment, a damaged PSI receptor, or denatured and degraded photosynthetic pigment–protein complexes [56,57].

Parameters extracted from the OJIP curve, such as $\varphi_{Ro}$, $\gamma_{RC}$, ABS/RC, and ABS/$CS_M$, can provide detailed information on the photosynthetic process [33,34]. The $\varphi_{Ro}$ and $\gamma_{RC}$ values decreased in H + W and were lower than those in the other three treatments (Table 3), indicating a decrease in the quantum yield of the electron transfer from $Q_A^-$ to the reduction of the end electron acceptors on the acceptor side of PSI under the heat stress treatment with distilled water [44,45,58]. Previous studies reported that the decrease in ABS/$CS_M$, $TR_0$/$CS_M$, and $ET_0$/$CS_M$ reflects the active reaction centers (RCs) being converted into inactive RCs, reducing the efficiency of trapping and causing a decline in PS II activity [53,59]. ABS/$CS_M$, RC/$CS_M$, $TR_0$/$CS_M$, and $ET_0$/$CS_M$ decreased and $DI_0$/$CS_M$ increased in the leaves of H + W compared to H + $Ca^{2+}$ (Figure 5), indicating that both the size of the functional antenna and the exciton-specific rate captured by the open $RC_S$ were reduced [60,61]. The maximum photochemical quantum yield ($F_v/F_m$) is mainly used as a chlorophyll a fluorescence indicator of heat stress [62]. The $F_v/F_m$ usually decreases dramatically after heat treatment [27,63]. PI refers to the energy conservation between photons absorbed by PS II and the reduction of intersystem electron acceptors ($PI_{ABS}$) [44]. In our study, the $F_v/F_m$ of the H + W treatment was significantly lower than that in CK + W (Table 3); this is consistent with previous research [63]. The $F_v/F_m$ and $PI_{ABS}$ of the H + W treatment were both lower than those in H + $Ca^{2+}$ (Table 3). However, the $F_v/F_m$ and $PI_{ABS}$ parameters in the two treatments with CaCl₂ (CK + $Ca^{2+}$ and H + $Ca^{2+}$) were not different from those in the control. In addition, the $\Delta_{Vt}$ curve showed that the ΔK, ΔJ, and ΔI of H + W were higher than those in the other three treatments (Figure 4B). Previous studies found that DF is associated with the recombination of S2 or S3 states of the oxygen evolving complex (OEC) with $Q^-_A$ or $Q^-_B$ [63], $I_1$ and $I_2$ points of DF are most likely related to the electron transfer from the $Q_B$ to $PQH_2$ of PS II and re-oxidized by PS I [62]. The $I_1$ point of DF appears in the J-I phase of PF dynamics and at the end of the fast phase of MR, whereas the $I_2$ point of DF appears in the I-P phase of PF and the slow phase of MR. At the $I_1$ and $I_2$ points of DF, the H + W curve is different from all of the other curves, illustrating that the electron transfer ability of PS I and PS II was inhibited in the leaves of *R. × pulchrum* under heat stress. This could be alleviated by using exogenous CaCl₂. These results illustrate that exogenous CaCl₂ can improve the heat tolerance of *R. × pulchrum* [27,44,64,65], suggesting that PSII and PSI were damaged and that the ability to pass electrons downstream (electron transfer from $Q_A$ to $Q_B$) was inhibited in the leaves of *R. × pulchrum* under heat stress whereas this damage could be alleviated by using exogenous CaCl₂ [44,66].

## 5. Conclusions

This study aimed to explore the effect of exogenous $CaCl_2$ on the leaf microstructure, leaf epidermal ultrastructure, and chlorophyll a fluorescence of *R. × pulchrum* under heat stress. The results of this study revealed that exogenous $CaCl_2$ improves the heat tolerance ability of *R. × pulchrum* by regulating the behavior of its epidermal cells and stomata, which, in turn, changes the arrangement and number of palisade tissue cell layer in its leaves. Meanwhile, under heat stress, its electron transfer ability from $Q_A$ to $Q_B$ was inhibited which could be alleviated by using exogenous $CaCl_2$. The results found that exogenous $CaCl_2$ might alleviate the inhibitory impact of heat stress on the anatomical and fast chlorophyll fluorescence features of *R. × pulchrum*. Our study could help in the understanding of the role of exogenous $CaCl_2$ in regulating the heat resistance of plants.

**Author Contributions:** Conceptualization, J.S. and S.J.; methodology, X.L., X.P. and Y.H.; software, X.P. and Y.H.; resources, S.J. and J.S. Shen; data curation, S.J. and J.S.; writing—original draft preparation and writing—review and editing, H.C., S.J. and J.S.; funding acquisition, S.J. All authors have read and agreed to the published version of the manuscript.

**Funding:** This research was funded by the National Key Research and Development Project (2019YFE0118900), the National Natural Science Foundation of China (31971641, 32201608), the Zhejiang Provincial Natural Science Foundation of China (LY16C160011), and the Jiyang College of Zhejiang A&F University under grant (RQ1911B07).

**Data Availability Statement:** Not applicable.

**Conflicts of Interest:** The authors declare no conflict of interest.

## Abbreviations

$F_v/F_m$, maximal quantum efficiency of PSII; $PI_{ABS}$, the performance index on an absorption basis; $\psi_{Eo}$, a trapped exciton moves an electron into the electron transport chain beyond $Q_A^-$ at $t = F_o$; $\delta_{Ro}$, efficiency required for an electron to be transferred from reduced carriers between the two photosystems to the PSI end acceptors; $\gamma_{RC}$, a given chlorophyll *a* molecule that functions as the PSII reaction center; $\varphi_{Eo}$, quantum efficiency of electron transfer at $t = F_o$; $\varphi_{Ro}$, quantum yield for reduction of the terminal electron acceptors on the PSI acceptor side; $\varphi_{Do}$, quantum yield for energy dissipation at $t = F_o$; $ABS/RC$, absorption flux per reaction centre (RC) basis of membrane model; $TR_0/RC$, trapped energy flux per RC (at $t = 0$); $DI_0/RC$, dissipation energy flux per RC (at $t = 0$); $ET_0/RC$, electron transport flux per RC (at $t = 0$); $t_{FM}$, time (in ms) to reach the maximal fluorescence intensity $F_M$; $ABS/CS_M$, absorption flux per excited leaf cross-section (CS) of leaf model; $RC/CS_M$, reaction centre per CS (at $t = t_{FM}$); $TR_0/CS_M$, trapped energy flux per CS (at $t = t_{FM}$); $ET_0/CS_M$, electron transport flux per CS (at $t = t_{FM}$); $DI_0/CS_M$, dissipation energy flux per CS (at $t = t_{FM}$).

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
