# Peer review of "Beneficial Effect of Exogenously Applied Calcium Chloride on the Anatomy and Fast Chlorophyll Fluorescence in Rhododendron × pulchrum Leaves Following Short-Term Heat Stress Treatment"

_agronomy, doi:10.3390/agronomy12123226_

Round 1
Reviewer 1 Report
The manuscript of Shen et al. deals with the effect of calcium chloride on the anatomy and photosynthetic electrons transport of heat stressed Rhododendron × pulchrum leaves.
The main conclusion of the work is that CaCl2 treatment provides some protection against heat-induced stress in rhododendron, similarly to other plants, which was reported earlier. This might be interesting for practical purposes considering the scope of the journal. However, the manuscript does not focus on that aspect, instead it attempts to provide some conclusion on the mechanism of the CaCl2 effect based on half-digested and unexplained parameters of photosynthetic electron transport. The beneficial effect of CaCl2 treatment is shown only after two days of heat stress, but does not demonstrated for longer period. In the present form the manuscript has very little use for the readership of the Agronomy journal.
Problems:
1, The CaCl2 treatment induces a large (ca. 2x) increase of stomata density in the heat treated leaves (Fig. 3. E vs M), and most of the stomata in the H + Ca2+ leaves appears to be open (Fig. 3M and Table 3). This is quite unexpected, why would it be beneficial for the plants to have a large number of open stomata under heat stress, which would significantly enhance water loss?
2, The Discussion contains many details which should be presented in the Results (description of data shown in the figures and the tables). On the other hand very little information is provided about the meaning and background of the photosynthetic parameters. This is especially true for the delayed fluorescence and modulated 820 nm reflection data. These are rather exotic measurements, and I doubt that they provide useful information for the readers of the Agronomy journal. What is the meaning of the D0, I1, I2 and D2 points on the DF curve? What is happening at the I1 point? Why are we happy that the H+W curve is different (at the I1 point) from all the other curves? The same question is valid for the MR/MR0 data. What does the drop of MR/MR0 ratio at 10 ms mean? Similar problems arise for the much more widely used Chl fluorescence parameters. What is the meaning and importance of the relative changes in K, J, I and P points of the OJIP curve (Fig. 4B). These might be familiar for specialists of photosynthetic electron transport studies, and could be dig out from the literature with lots of work, but provide very little information for the readership of Agronomy.
3, In Fig. 5 it is not defined which RC is depicted (PSI of PSII)?
4, The text should be checked for proper English use.
Minor:
The word „Title” is not needed in the title.
In the legend to Fig. 1. the Euro sign should be replaced with E.
In Table 1. the a, b, c letters which indicate statistical significance are in a wrong place with the exception of the first column (Upper epidermis).
l. 156: the second CK+W is most most likely H+W.
l. 235. „2” should be in subscript in CaCl2
l. 301. Is a left-over from the instruction to the authors and should be deleted.
It is unclear how much was applied from the CaCl2 solutions in the foliar spraying treatments. The Materials and Methods section specifies only the molar concentration, but not the volume which was sprayed on the leaves.
Author Response
We have carefully read all the comments, according to the suggestions we corrected the content in the main text. Thanks for your suggestion and help.
1, The CaCl2 treatment induces a large (ca. 2x) increase of stomata density in the heat treated leaves (Fig. 3. E vs M), and most of the stomata in the H + Ca2+ leaves appears to be open (Fig. 3M and Table 3). This is quite unexpected, why would it be beneficial for the plants to have a large number of open stomata under heat stress, which would significantly enhance water loss?
Response: Our results found ‘The CaCl2 treatment induces a large increase of stomata density in the heat treated leaves’. This result is consistent with previous study that calcium ion influx was predominant in the guard cells. Compared to that under control conditions, the stomatal density under stress conditions increased significantly (Zhu et al., 2019).
To avoid drought stress, soil water content was kept stability during the heat stress experiment. Most of the stomata in the H + Ca2+ leaves appear to be open (Fig. 3M and Table 3). This maybe through transpiration, accelerates water transport and increases the heat tolerance of plants.
References
Zhu, X.; Wang, L.; Yang, R.; Han, Y. Y.; Hao, J. H.; Liu, C. J.; et al. Effects of exogenous putrescine on the ultrastructure of and calcium ion flow rate in lettuce leaf epidermal cells under drought stress. Hortic. Environ. Biote. 2019, 60, 479-490. doi: 10.1007/s13580-019-00151-7
2, (1) The Discussion contains many details which should be presented in the Results (description of data shown in the figures and the tables).
Response: The description of data shown in the figures and the tables contains in Discussion had been added in Results part. (Line 140-141, Line 164, Line 178-183, Line 197-198, Line 215-217)
(2) On the other hand very little information is provided about the meaning and background of the photosynthetic parameters. This is especially true for the delayed fluorescence and modulated 820 nm reflection data. These are rather exotic measurements, and I doubt that they provide useful information for the readers of the Agronomy journal. These might be familiar for specialists of photosynthetic electron transport studies, and could be dig out from the literature with lots of work, but provide very little information for the readership of Agronomy. What is the meaning of the D0, I1, I2 and D2 points on the DF curve? What is happening at the I1 point? Why are we happy that the H+W curve is different (at the I1 point) from all the other curves? The same question is valid for the MR/MR0 data. What does the drop of MR/MR0 ratio at 10 ms mean? Similar problems arise for the much more widely used Chl fluorescence parameters. What is the meaning and importance of the relative changes in K, J, I and P points of the OJIP curve (Fig. 4B).
Response: The meaning of Chl fluorescence parameters of PF, DF and MR/MR0 had been added in Introduction part (Line 62-65). The information had been provided about the meaning and background of the photosynthetic parameters, including delayed fluorescence and modulated 820 nm reflection data (Line 272-273).
Recently, the familiar study titled ‘Changes in Photosynthetic Characteristics of Paeonia suffruticosa under High Temperature Stress’ published in Agronomy. (https://doi.org/10.3390/agronomy12051203)
The H+W curve is different (at the I1 point) from all the other curves illustrate the ability of electrons transfer in PS II was inhibited in the leaves of R. × pulchrum under heat stress, which could be alleviated by using exogenous CaCl2.
3, In Fig. 5 it is not defined which RC is depicted (PSI of PSII)?
Response: We have defined ‘RC’ in Figure 5 legend. Figure 5. Energy flux models in photosystem II for the leaf samples of Rhododendron × pulchrum under four treatments.
4, The text should be checked for proper English use.
Response: Our language has been reviewed by the MDPI (https://www.mdpi.com/authors/english.). Here is the certification:
Minor:
- The word “Title” is not needed in the title.
Response: We have deleted the word “Title” in the title. (Line 2)
- In the legend to Fig. 1. the Euro sign should be replaced with E.
Response: We have modified “the Euro sign” to E. (Line 145)
- In Table 1. the a, b, c letters which indicate statistical significance are in a wrong place with the exception of the first column (Upper epidermis).
Response: We have modified the place of the a, b, c letters in Table 1. (Line 152)
- 156: the second CK+W is most likely H+W.
Response: We have changed “CK+W” to “H+W”. (Line 161)
- 235. „2” should be in subscript in CaCl2
Response: We have changed “CaCl2” to “CaCl2”. (Line 239)
- 301. Is a left-over from the instruction to the authors and should be deleted.
Response: We have deleted this sentence. (Line 314)
- It is unclear how much was applied from the CaCl2 solutions in the foliar spraying treatments. The Materials and Methods section specifies only the molar concentration, but not the volume which was sprayed on the leaves.
Response: About 10 ml per plant, evenly spray both sides of the leaves. (Line 88)

Reviewer 2 Report
The article “Beneficial effect of exogenously applied calcium chloride on the anatomy and fast chlorophyll fluorescence in Rhododendron × pulchrum leaves following short-term heat stress treatment by Shen et al is related to understand the effect of calcium on heat response in Rhododendron, a very sensitive genus to heat of ornamental plants greatly appreciated worldwide.
The introduction clearly stated the relevance of this study.
Material and Methods: There are some points that have to been clarified by the authors:
The supplemental table 1 was not available.
Why dehydrated samples were used for microestructure and ultraestructure analyses?
The statistical analysis is not clearly explained.
Results: There are some points that have to been clarified.
Figure 1: Plant images in E and F are the same.
Figure 2.- The images are not very clear. It will be easy to understand the differences if the different cell types are identified.
Table 1, table 2 and table 3. The numbers 1-5 inside the table are not necessary. For example, you can use CK + W instead of 1.
Figure 3. The quality of these images has to be improved and the size bar has to be included in all images.
Figure 5. The quality of these images has to be improved.
Discussion: The discussion has to be improved. re 5. The quality of these images has to be improved.
Discussion: The discussion has to be improved.
Author Response
- Material and Methods: There are some points that have to been clarified by the authors:
(1) The supplemental table 1 was not available.
Response: Supplement Table 1 as follows:
Supplement Table 1 The effect of different concentrations of CaCl2
Treatment |
concentration(mmol/L) |
Proportion of exhibiting heat damage after 3 days of heat stress treatment(%) |
CK |
0 |
80±14 a1 |
CaCl2 |
5 |
44±9 bc |
10 |
24±9 d |
|
20 |
28±11 cd |
|
30 |
60±14 b |
1 Mean ± SE, followed by the same letter in the same column, indicates no significant difference at the 0.05 level, based on Duncan’s test
(2) Why dehydrated samples were used for microestructure and ultraestructure analyses?
Response: The semithin sections and ultrathin sections methods were used for microestructure and ultraestructure analyses. Dehydrated samples are usually used in the experimental process of the two conventional methods. See the references below:
[1] Zhang, S.; Jiang, H.; Peng, S. M.; Korpelainen, H.; Li, C.Y. Sex-related differences in morphological, physiological, and ultrastructural responses of Populus cathayana to chilling. J. Expt. Bot. 2011, 62: 675-686. doi: 10.1093/jxb/erq306
[2] Ruppel, N. J.; Logsdon, C. A.; Whippo, C.W. A mutation in Arabidopsis seedling plastid development affects plastid differentiation in embryo-derived tissues during seedling growth. Plant Physiol. 2011, 155, 342-353. doi: 10.1104/pp.110.161414
(3) The statistical analysis is not clearly explained.
Response: We had added ‘2.4 Statistical analysis’ part, as follows:
The experimental design was a completely randomized design. The data were analyzed using SPSS 18.0 software (SPSS Inc. Chicago, IL, USA), and one-way ANOVA and Duncan’s multiple range test was performed for the data variable, with the significance level set at the default α = 0.05.
- Results: There are some points that have to been clarified.
(1) Figure 1: Plant images in E and F are the same.
Response: In figure 1, plant images in E and F are the same plant under different processing time.
(2) Figure 2.- The images are not very clear. It will be easy to understand the differences if the different cell types are identified.
Response: The quality of these images had been improved and had marked the cells type in Figure 2.
(3) Table 1, table 2 and table 3. The numbers 1-5 inside the table are not necessary. For example, you can use CK + W instead of 1.
Response: The numbers 1-5 inside the table had been deleted in Table 1, Table 2 and Table 3.
(4) Figure 3. The quality of these images has to be improved and the size bar has to be included in all images.
Response: The quality of these images had been improved, and the size bar are included in all figures. Please check it.
(5) Figure 5. The quality of these images has to be improved.
Response: The quality of these images had been improved. The figures will be automatically compressed when put in word. We provide the figures separately, please check it.
- Discussion: The discussion has to be improved.
Response: The discussion had been modified in manuscript using ‘Modify Mode’.

Round 2
Reviewer 1 Report
The manuscript was revised and the minor points of my review were taken into account.
It is still unclear why and how could CaCl2 protect against heat stress by inducing the open state of stomata, which should lead to increased water loss, a highly undesired effect, under elevated temperatures.
The Discussion still contains lot's of details which belong to the Introduction and Results, and mechanistically it is completely unclear why heat stress would modify the electron transport rate at the level of the quinone electron acceptors in PSII, and how could CaCl2 reverse it.
Heat stress studies in isolated thylakoid and PSII preparations identified the water-oxidizing complex at the donor side of PSII as the primary heat sensitive site, and it is highly unlikely that the situation would be different in intact leaves. I am afraid that the problems regarding the unsatisfactory clarification of the mechanism of the heat-induced effects in PSII and its amelioration by CaCl2 stem from the application of the half (or less) understood measurement methods (DL and MR).
However, the scope of Agronomy is not the clarification of mechanisms, therefore the presentation of observations could be a sufficient justification to publish it at the level of this journal.
Author Response
Response:
Thank you for your comments. These comments are valuable and helpful for revising and improving our paper.
Our study presented the observations of the exogenous CaCl2 effect the leaf anatomical structure and the behavior of epidermal cells, stomata and electron transport of photosystem in leaves under heat stress. CaCl2 protect against heat stress by inducing the open state of stomata, which may be due to the fact that the soil field water-holding capacity is sufficient during the treatment process. The mechanism that CaCl2 protect against heat stress by regulating the behavior of epidermal cells, stomata and electron transport of photosystem in leaves need to be further studied.
In the revised paper, some of the contents in the discussion have been moved to the introduction. (Line 63-73)
Chlorophyll fluorescence parameters reflect the electron transfer in PSI, PSII, and photosynthetic electron carriers in the photosystem. Based on results obtained during ChFl measurements, DF is associated with the recombination of S2 or S3 states of the oxygen evolving complex (OEC) with Q-A or Q-B, I1 and I2 points of DF are most likely related to the electron transfer from the QB to PQH2 of PS II and re-oxidized by PS I. We have modified the Discussion part using “Track Changes” function. (Line 302-305; 326-331)